# Socio-Demographic, Environmental, and Clinical Factors Influencing Diabetes Mellitus Control in Community Pharmacies of Lahore Pakistan

**DOI:** 10.3390/healthcare13212733

**Published:** 2025-10-28

**Authors:** Seerat Shahzad, Muhammad Zahid Iqbal, Naeem Mubarak, Tahneem Yaseen, Khalid M. Orayj, Saad S. Alqahtani

**Affiliations:** 1Department of Pharmacy Practice, Faculty of Pharmaceutical Sciences, Lahore University of Biological & Applied Sciences, Lahore 53400, Pakistan; seerat.shahzad@ubas.edu.pk (S.S.); naeem.mubarak@ubas.edu.pk (N.M.); tahneem.yaseen@ubas.edu.pk (T.Y.); 2Department of Clinical Pharmacy, College of Pharmacy, King Khalid University, Abha 61421, Saudi Arabia; korayg@kku.edu.sa (K.M.O.); ss.alqahtani@kku.edu.sa (S.S.A.)

**Keywords:** demographics, factor, diabetes mellitus, confounders, predictors, pharmacy, pharmacist

## Abstract

**Background**: Diabetes Mellitus (DM) represents a significant public health challenge in Pakistan, with a high prevalence exacerbated by various socio-demographic, clinical, and environmental factors. Community pharmacies offer an accessible setting for managing chronic diseases, yet the combined influence of these factors on diabetes control within Pakistani community settings remains underexplored. **Objective**: This study aimed to assess the impact of socio-demographic, environmental, and clinical factors on diabetes control among patients attending community pharmacies in Lahore, Pakistan. **Methods**: A cross-sectional study was conducted involving 321 patients with type 2 diabetes recruited from community pharmacies across three regions of Lahore. A structured questionnaire, developed based on international guidelines, was used to collect data on socio-demographic characteristics, clinical history, lifestyle behaviors, and environmental factors. Diabetes control was categorized as controlled, partially controlled, or uncontrolled. Data were analyzed using descriptive statistics, chi-square tests, and multiple logistic regression in SPSS version 26.0. **Results**: Key socio-demographic predictors of better diabetes control included higher education levels (AOR = 1.317–2.338, *p* ≤ 0.006) and non-obese status (AOR = 1.057, *p* = 0.006). Significant clinical and lifestyle predictors were treatment adherence (AOR = 1.287, *p* < 0.001), regular physical activity (AOR = 1.387, *p* < 0.001), healthy dietary patterns (AOR = 1.317, *p* < 0.001), and longer duration of diabetes (>5 years, AOR = 1.277, *p* = 0.008). Conversely, a family history of diabetes (AOR = 1.967, *p* < 0.001) and the presence of comorbidities were associated with poorer control. Rural residence showed lower odds of good diabetes control (AOR = 0.857, *p* = 0.001). Smoking status was also influential, with ex-smokers demonstrating better control than current smokers. **Conclusions**: Diabetes control is multifactorial, strongly influenced by education, residence, obesity, lifestyle behaviors, and treatment adherence. Interventions targeting modifiable risk factors through patient education, lifestyle counseling, and personalized care are essential to improve diabetes outcomes in community settings. These findings underscore the critical role of community pharmacists in providing holistic diabetes management.

## 1. Introduction

One of the biggest global public health concerns is diabetes mellitus (DM) [1]. About 537 million people worldwide received a diabetes diagnosis in 2021, and research suggests that by 2045, that figure might increase to 783 million [2]. The rising frequency of diabetes results from several interacting factors. These include obesity, changing lifestyles, population aging, and rapid urbanization [3,4].

In Pakistan, the prevalence of diabetes is extremely concerning [5]. A 2021 International Diabetes Federation (IDF) survey found that Pakistan has a higher-than-average diabetes rate of 9.6% [6]. Accordingly, Pakistan has one of the highest global diabetes rates [7]. Numerous socio-demographic factors, including poor self-care practices, a lack of knowledge, and low health literacy, contribute to this significant rise in cases [8].

Many risk factors contribute to the prevalence and progression of diabetes mellitus [9]. Patients with hypertension or dyslipidemia, as well as those who have previously had pregnancy-induced diabetes, are more likely to have this disease [10]. Additionally, there is strong evidence that type 2 diabetes is a hereditary risk [11]. In relation to Type 2 diabetes, several non-genetic variables have been identified [12].

Obesity and inactivity are important modifiable risk factors because they contribute significantly to insulin resistance, as do excess body weight, especially central obesity, and a sedentary lifestyle [13]. A diet high in carbohydrates has been linked to a five-fold-increased risk of Type 2 diabetes [14]. Male gender and advancing age have also been associated with an increased incidence of type 2 diabetes [15].

Concomitant conditions such as obesity, heart disease, chronic renal disease, hypertension, and hyperlipidemia are frequently experienced by people with diabetes [16]. These comorbidities not only complicate the therapy but also increase the risk of mortality [17]. Additionally, in the treatment of diabetes, psychological factors such as depression and anxiety may negatively affect glycemia control [18]. Environmental factors such as pollution exposure and urbanization have been linked to insulin resistance [19].

Research has demonstrated that a higher incidence of type 2 diabetes is associated with residential noise, air pollution, and socioeconomic deprivation at the level of the community [20]. Opportunities for physical activity tend to be only a few in urban settings [21] and easier access to unhealthy food options [22], increasing the prevalence of obesity and, consequently, diabetes [23]. Poor socioeconomic level, a lack of education, and restricted access to medical facilities are examples of socio-demographic characteristics that are important in the onset and treatment of disease [24]. To properly control diabetes, a multimodal strategy is needed [25]. Basic strategies include modifications in lifestyle, such as weight control, frequent exercise, and a balanced diet [26]. Glycemia in patients with type 2 diabetes mellitus has been shown to improve with treatment with diet alone, insulin, sulfonylurea, or metformin [27].

Regular blood-glucose monitoring and self-management education are vital to prevent diabetes complications [28,29]. Community pharmacies, due to their accessibility, can provide health education, guidance on lifestyle changes, and medication counseling [30]. Pharmacists, therefore, play a central role in diabetes care, as they help in recognizing vulnerable patients, encourage adherence to therapy, and collaborate with other healthcare providers [31,32].

Despite the high burden of diabetes in Pakistan, only a few investigations have examined how the combined influence of socio-demographic, clinical, and environmental factors affects the management of diabetes, particularly within community settings. Most prior research has been conducted on hospital-based populations or limited to any biological outcome, thereby reducing the generalizability of their conclusions. Addressing this gap, the present study aims to assess the impact of diverse confounding variables on diabetes control among patients attending community pharmacies in Lahore. By examining an outpatient population, this research provides broader insights into real-world management and aims to identify practical solutions for improving diabetes control at the community level.

## 2. Methodology

The present study employed a cross-sectional observational design to assess how socio-demographic, clinical, and environmental factors influence diabetes control in patients attending community pharmacies in Lahore, Pakistan. This city was purposely chosen because of its high prevalence of diabetes and socioeconomically diverse population. To strengthen external validity, sampling was extended to several urban and suburban regions.

A validated, structured questionnaire was designed in accordance with international standards, specifically the NICE guidelines, the International Diabetes Federation recommendations, and the American Diabetes Association’s standards of medical care. The survey was divided into two main domains. The first domain obtained socio-demographic information including age, gender, education, occupation, income, and family history of diabetes. The second domain evaluated clinical, environmental, and lifestyle factors linked with disease control, including duration of diabetes, present glycemic control, medication adherence, dietary and physical practices, smoking status, comorbidities such as hypertension, CVS diseases, and hyperlipidemia, as well as environmental exposures such as air pollution, neighborhood condition, healthcare accessibility, and recreational opportunities.

No formal a priori power calculation was performed because recruitment depended on the number of eligible patients visiting community pharmacies during the study period. The final sample (*n* = 321) yields a two-sided 95% confidence interval half-width of ≈±5% for proportions near 0.5, ensuring adequate descriptive precision. Post hoc power analysis using G*Power 3.1 for χ^2^ tests (df = 4, α = 0.05) indicated that this sample provides approximately 80–96% power to detect the effect sizes actually observed in our data (Cramer’s V = 0.22–0.30, corresponding to η^2^ ≈ 0.05–0.09). These results demonstrate that the achieved sample size was sufficient for reliable inference within the exploratory design.

Research activities were carried out in three regions of Lahore: Gulberg, Garhi Shahu and Harbanspura. Before data collection, community pharmacists were contacted, and their cooperation was obtained. Convenience sampling was used to recruit patients, particularly those who frequently visited community pharmacies for diabetic prescriptions or counseling. Convenience sampling was used due to the practical constraints of recruiting patients in community pharmacy settings. While this approach facilitated access to a diverse pool of participants, it may have introduced several potential biases. The primary concern is selection bias, as patients who visit pharmacies frequently or are more adherent to therapy are more likely to be enrolled, possibly leading to overrepresentation of individuals with higher health awareness. This may in turn affect the observed relationships between adherence, lifestyle, and diabetes control. The sample may also lack representativeness of the broader diabetic population, particularly those managed exclusively in hospital or rural settings. Additionally, response bias could occur if participants provided socially desirable answers during interviews. To mitigate these limitations, data were collected from multiple pharmacy sites located in socioeconomically distinct regions of Lahore, and standardized data-collection procedures were followed to maintain consistency and minimize interviewer bias. Before enrollment, the study’s purpose and process were explained, privacy protections were emphasized, and written informed consent was obtained from all participants.

In total, 321 individuals with diabetes were recruited, and each of them provided voluntary consent to join the study. Data were collected through structured, face-to-face interviews conducted in the pharmacy setting. Only trained researchers conducted the interviews to enhance reliability and to minimize interviewer influence. Participant privacy was preserved by maintaining strict confidentiality and anonymity at all stages.

Ethical approval for this study was granted by the Research and Ethics Review Committee of Lahore University of Biological and Applied Sciences (Approval Code: UBAS/ERB/FoPS/25/003) on 25 July 2025. Participant autonomy and confidentiality were protected through strict compliance with the ethical standards established by the committee.

### 2.1. Statistical Analysis

Statistical analysis was performed with SPSS software version 26.0. Descriptive statistics were used to outline patients’ demographics and possible confounding factors. Associations between the socio-demographic, environmental, and clinical variables and diabetes outcomes were examined by Chi-square tests, considering *p*-value < 0.005 as statistically significant. Effect sizes were measured using Phi and Cramer’s V. To further evaluate the independent influence of clinical, socio-demographic, and environmental factors on diabetes control, Regression analysis was also applied while controlling relevant confounders. For logistic regression analyses, the dependent variable was dichotomized as Good Diabetes Control (coded = 1) versus Poor Diabetes Control (coded = 0), where the latter combined both partially controlled and uncontrolled categories. An Adjusted Odds Ratio (AOR) greater than 1 therefore indicates a higher likelihood of achieving good control, whereas an AOR less than 1 indicates reduced odds of good control.

### 2.2. Operational Definition of Diabetes Control

Glycemic control status was determined using the most recent HbA1c value, as reported by the participant based on their latest laboratory results or physician consultation. Classification followed the cut-off values endorsed by the Pakistan Endocrine Society (PES) and aligned with international guidelines (ADA, IDF, NICE):Controlled diabetes: HbA1c < 7.0%Partially controlled diabetes: HbA1c between 7.0% and 8.0%Uncontrolled diabetes: HbA1c > 8.0%

These thresholds reflect the standard glycemic targets for adults with type 2 diabetes in Pakistan, as recommended in national and international clinical practice guidelines. Participants who could recall their physician-reported HbA1c or provide laboratory documentation were classified accordingly.

## 3. Results

Participants showed diverse socio-demographic, environmental, and clinical profiles. A total of 321 patients with type 2 diabetes mellitus (T2DM) were included in the study. Of the 321 respondents, 47.0% (*n* = 151) were male and 53.0% (*n* = 170) were female. Age distribution showed that 27.2% were 18–40 years, 36.4% were 41–65 years, and an equal proportion (36.4%) were over 65 years. Occupationally, 35.5% were retired, 28.7% were employed in sedentary jobs, 17.8% were unemployed, and 18.1% were engaged in physically active work.

Educational attainment varied, with 34.3% having no formal education, 24.9% reaching secondary level, and only 3.1% and 2.5% holding graduate and postgraduate degrees, respectively. With respect to marital status, 41.7% were single, 27.7% were married, and 28.0% were divorced. More than half (57.9%) reported having only one child.

Residence was almost evenly split between urban (49.5%) and rural (50.5%) areas. Over half of the participants lived in rented accommodation (58.6%), and 63.6% belonged to joint family systems. Monthly income varied, with 34.6% earning 20–50k and 22.1% earning more than 100k.

Over half of the study sample (54.5%) were classified as obese, and 40.8% had type 2 diabetes for longer than five years. A family history of diabetes was present in 55.1% of cases. The prevalence of comorbidities was considerable, with hypertension (46.4%), dyslipidemia (49.8%), renal diseases (45.8%), cardiovascular disease (49.2%), and non-alcoholic fatty liver disease (45.8%). Among female participants, 26.3% reported polycystic ovary syndrome.

With regard to lifestyle, 42.7% were current smokers, while 35.2% had never smoked. More than half (53.3%) reported engaging in recommended levels of physical activity (≥150 min/week), though 53.6% had unhealthy dietary patterns. Notably, 57.3% of participants did not adhere to their treatment plan. More details regarding the demographic characteristics can be obtained from Table 1 as follows:

Multiple logistic regression was performed to evaluate the impact of socio-demographic and clinical variables on diabetes control. The findings revealed that a number of factors contributed significantly to the effective management of diabetes mellitus.

### 3.1. Socio-Demographic Predictors

Socio-demographic factors significantly influence diabetes control, including occupation, residence, education, and obesity. Univariate analyses revealed that individuals with sedentary occupations had better control compared to those who are retired or unemployed. Higher education (secondary or graduate) was positively associated with diabetes control compared to no formal education (*p* < 0.001; η^2^ = 0.06–0.07). Rural residence was associated with reduced odds of achieving good diabetes control compared with urban residence. Obesity was negatively associated with diabetes control, with non-obese participants more likely to achieve adequate regulation (*p* < 0.001, η^2^ = 0.02).

Multivariate analyses validated the independent effects of residence, education, and obesity, while gender, marital status, and family size did not reach statistical significance in the adjusted models. Detailed findings are presented in Table 2.

### 3.2. Environmental Predictors

Environmental exposures such as neighborhood condition, air pollution, healthcare accessibility, and recreational opportunities were included in the data collection form. However, due to incomplete responses for these variables, they were not included in the final statistical analysis to avoid biased interpretation and ensure data validity.

### 3.3. Clinical Predictors

As presented in Table 3, several lifestyle and clinical factors were significantly associated with diabetes control. Univariate analyses showed that longer duration of diabetes (>5 years), absence of hypertension, renal or cardiovascular disease, and in the univariate analysis, a positive family history showed a statistically significant association with glycemic control (*p* < 0.001); however, this relationship reversed direction in the multivariate model, where family history was associated with poorer glycemic control. Regular physical activity (>150 min/week), healthier dietary patterns, and adherence to treatment demonstrated significantly better outcomes (*p* < 0.001 for all), with small to medium effect sizes (η^2^ = 0.08–0.09).

In contrast, poor diet, physical inactivity, treatment non-adherence, and comorbidities such as NAFLD, hypertension, and dyslipidemia demonstrated poor diabetes control. Smoking status also influenced outcomes, with ex-smokers achieving superior control compared to current or never-smokers (*p* < 0.001, η^2^ = 0.08).

Multivariate analyses confirmed the independent significance of physical activity, treatment adherence, diet, and smoking status, highlighting their pivotal role in diabetes management beyond clinical comorbidities.

Table 2 and Table 3 present the crude and adjusted odds ratios (ORs) for evaluating the association between various confounders and control in diabetes.

Table 2 summarizes socio-demographic predictors influencing diabetes control among patients visiting community pharmacies in Lahore, Pakistan. Significant predictors included education, residence, obesity, and occupation. Those with higher education (AOR = 1.378, *p* = 0.006, η^2^ = 0.06) had improved outcomes, and non-obese participants were also more likely to achieve control (AOR = 1.057, *p* = 0.006, η^2^ = 0.02).

Table 3 indicates clinical and lifestyle factors influencing diabetes control. Better diabetes control was strongly associated with treatment adherence (AOR = 1.287, *p* < 0.001, η^2^ = 0.09), regular exercise (AOR = 1.387, *p* < 0.001, η^2^ = 0.09), and a healthy diet (AOR = 1.317, *p* < 0.001, η^2^ = 0.08). Longer disease duration (AOR = 1.277, *p* = 0.008) also showed a significant association with improved diabetes control. Consistent with poorer outcomes, a positive family history of diabetes was associated with significantly higher odds of poor glycemic control (AOR = 1.967, *p* < 0.001, η^2^ = 0.09). Furthermore, comorbidities such as hypertension, dyslipidemia, renal disease, CVD, and NAFLD reduced outcomes. Smoking status also influenced results, with ex-smokers (AOR = 0.667, *p* < 0.001, η^2^ = 0.08) having better glycemic control compared to both current smokers and never smokers.

## 4. Discussion

This cross-sectional study conducted in community pharmacies of Lahore, Pakistan, examined socio-demographic, clinical, and environmental predictors of diabetes control. As of socio-demographic factors, the findings indicate that the education level and occupation type are significant predictors of diabetes control. Occupation type was the most significant predictor. Sedentary-occupation participants reported the highest level of diabetes control (53.4%), which is greater than that of the unemployed or retired groups (AOR = 1.208. *p* = 0.047). More financial stability is frequently associated with a sedentary, office-based job, which enhances access to healthcare, prescription drugs, and better food alternatives [33]. On the other hand, unemployment creates financial obstacles to managing diabetes, a problem that is prevalent in developing countries [34]. Participants who were retired had poor control, mostly due to age-related issues, polypharmacy, and comorbidities that make managing their diseases more difficult [35].

Additionally, there was a high correlation between diabetes management and education level. Participants with secondary and graduate education were more likely to have controlled diabetes compared to those with no formal education (AOR = 1.317, *p* = 0.001 for secondary; AOR = 2.338, *p* = 0.001 for graduates). This demonstrated the role of health literacy in effective self-management. Higher education provides people with the capacity to comprehend medical information, adhere to treatment, and adopt healthier lifestyle practices [36,37].

Diabetes results were also substantially impacted by marital status (*p* = 0.031), with married participants demonstrating greater control (44.9%) than those who were single or divorced, emphasizing the importance of spousal support in managing chronic diseases [38]. Although this conclusion is based on a fairly short sample size (*n* = 8), widowed participants also showed strong control. In this study, age (*p* = 0.187) and gender (*p* = 0.784) did not substantially correlate with diabetes management. Women and men had equal control rates, indicating similar challenges in healthcare in this context [15]. Across age groups, diabetes control remained uniformly difficult, though partial control was more common among those aged >65 years, likely reflecting cautious treatment strategies to avoid hypoglycemia [39].

Participants living in rented houses showed better control (49.8%) compared to participants who lived in their own homes (26.6%), a highly significant difference (*p* = 0.001). Owned houses often represent old, inherited properties in densely populated neighborhoods with restricted access to healthcare facilities [40]. In contrast, rented houses represent newer apartments in better-developed areas, with improved health literacy and access to resources [41]. In the bivariate analysis, a greater proportion of rural participants demonstrated good glycemic control (53.1%) compared with urban participants (22.6%; *p* < 0.001). However, after adjustment for potential confounders in the multivariate model, rural residence was associated with reduced odds of good diabetes control (AOR = 0.857, *p* = 0.001) compared with urban residence. Despite being closer to healthcare services, urban residents are at a disadvantage due to their sedentary lifestyle, high stress, and processed food consumption, whereas rural residents benefit from higher physical activity and a traditional diet [42,43].

Obesity significantly reduced diabetes control. Non-obese participants achieved more than double the rate of glycemic control (54.8%) compared to obese participants (24.0%) (*p* < 0.001). This shows obesity’s central role in insulin resistance and its position as the most important modifiable risk factor in type 2 diabetes [44]. Overweight and obesity significantly increases the risk of uncontrollable diabetes. Longitudinal analyses confirm adiposity as an independent predictor of both the onset and poor management of diabetes [45]. These findings prove the necessity of weight management as a central component of diabetes control strategies.

Monthly income showed no statistically significant linear association with diabetes control (*p* = 0.323). All income categories showed similar rates of control (30–42%), with the highest income (>100,000 PKR) indicating a greater percentage of uncontrolled diabetes (31.0%). This implies that knowledge and behavior can offset economic advantage, and that wealth by itself does not ensure efficient management [46].

Furthermore, the number of children a participant had was also not significantly linked with diabetes control. Parenting offers important social support, even if it can also cause financial and psychological burden. The lack of a clear correlation indicates that other, stronger clinical and behavioral factors have a greater impact than family size [47].

Although environmental factors were initially assessed, the number of complete responses for these items was insufficient for reliable analysis. Their exclusion from the final results was therefore due to missing data rather than lack of relevance, and future studies with larger and more complete datasets are recommended to further explore these associations.

The duration of the disease has a strong correlation with diabetes control among the clinical predictors. Those who had diabetes for more than five years were more likely to achieve control than those who had just received a diagnosis (AOR = 1.233, *p* = 0.008, η^2^ = 0.04). This research indicates that long-term patients are more involved with healthcare services, more used to treatment plans, and more capable of managing their own healthcare. However, this finding should be interpreted with caution. Rather than indicating a direct protective effect of disease duration, this relationship may reflect behavioral adaptation; in other words, individuals who have lived with diabetes for many years are often more familiar with their treatment regimens, attend regular medical consultations, and adhere more closely to dietary or pharmacological recommendations. Additionally, survivorship bias may partially explain this trend, as those who maintain follow-up and adhere to treatment are more likely to remain in the study population over time. Nevertheless, since most prior studies link longer duration with poorer glycemic outcomes due to progressive β-cell dysfunction, our finding should be viewed as exploratory and warrants further longitudinal validation. However, this is in contrast to a large body of research that associates inadequate management with a longer duration of disease because of greater complications [48]. Therefore, our results show that chronic patients in this group had better adherence.

A family history of diabetes was linked to worse control (AOR = 1.967, *p* < 0.001, η^2^ = 0.09), demonstrating the influence of genetic risk. Furthermore, comorbidities such as hypertension, cardiovascular disease, dyslipidemia, and renal disease were less likely to be under control since they exacerbate glycemic control and raise treatment problems for diabetes.

The control of diabetes was strongly and independently influenced by lifestyle variables. Regular exercisers (>150 min/week) had a substantially higher chance of achieving glycemic control (AOR = 1.387, *p* < 0.001, η^2^ = 0.09). This confirms the well-established results that exercise improves insulin sensitivity and decreases HbA_1_c levels. Dietary practices were also significant; those who followed healthy dietary practices managed their diabetes better than those who had poor dietary practices (AOR = 1.317, *p* < 0.001). Previous studies have shown that a Mediterranean or low-fat, low-sugar diet significantly improves metabolic outcomes [49].

Treatment adherence was one of the strongest predictors, and those who followed the treatment plan had noticeably better glycemic control (AOR = 1.287, *p* < 0.001, η^2^ = 0.09). This is consistent with global research demonstrating that non-adherence is one of the most common reasons for poor glycemic results despite medication treatment. Assessments of diabetes education programs have consistently shown that individuals with higher adherence levels had considerably lower HbA_1_c levels than those treated with standard care [50].

Diabetes results were further impacted by smoking status. Compared to present smokers, ex-smokers had superior glycemic control (AOR = 0.667, *p* < 0.001, η^2^ = 0.08). This finding is consistent with the evidence from Fukuoka Diabetes Registry, which documented dose- and time-dependent improvements in glycemic outcomes after smoking cessation [51]. Active smoking, on the other hand, makes insulin resistance worse and makes treatment more difficult.

In addition, the use of self-reported data collected through face-to-face interviews may have introduced recall bias and social desirability bias. Participants might have over-reported socially desirable behaviors such as regular medication adherence, healthy dietary habits, and physical activity, or under-reported less favorable behaviors. These biases could have led to an overestimation of positive lifestyle patterns and attenuated true associations between behavioral factors and glycemic control. Although the structured questionnaire and interviewer guidance helped improve accuracy, these inherent limitations should be considered when interpreting the results. Overall, the study emphasizes how lifestyle choices, clinical conditions, and socio-demographic traits all influence the multifactorial basis of diabetes control.

## 5. Conclusions

This study emphasizes the comprehensive impact of socio-demographic, clinical, and environmental factors influencing diabetes control in Pakistani community settings. Key predictors, such as education, residence, occupation, and housing status, were important social determinants, while obesity, family history of diabetes, comorbidities, and disease duration are crucial clinical factors influencing diabetes control. Lifestyle behaviors such as physical activity, healthy diet practices, smoking, and treatment adherence were among the most powerful determinants of glycemic control. Living in an urban region, smoking, being obese, and having comorbidities all made it difficult to treat diabetes; however, adherence, higher education, healthy eating, and regular exercise all improved diabetic outcomes. Improving education, counseling, and lifestyle habits can help address modifiable risks. A comprehensive strategy involving behavioral, educational, and customized therapy is needed to improve diabetes outcomes and quality of life. Future studies must examine these characteristics’ long-term consequences in order to develop sustainable diabetes care.

## Figures and Tables

**Table 1 healthcare-13-02733-t001:** Demographic details of study population (*n* = 321).

Demographics	N (%)
**Gender**	
Male	151 (47.0)
Female	170 (53.0)
**Age groups**	
18–40 years	87 (27.2)
41–65 years	117 (36.4)
More than 65 years	117 (36.4)
**Occupation Type**	
Sedentary	92 (28.7)
Active	58 (18.1)
Unemployed	57 (17.8)
Retired	114 (35.5)
**Education Level**	
No Formal	110 (34.3)
Primary	113 (35.2)
Secondary	80 (24.9)
Graduate	10 (3.1)
Postgraduate	8 (2.5)
**Marital Status**	
Single	134 (41.7)
Married	89 (27.7)
Divorced	90 (28.0)
Widowed	8 (2.5)
**Number of children**	
No child/NA	78 (24.3)
1 child only	186 (57.9)
2 children	20 (6.2)
More than 2	37 (11.5)
**Residence**	
Urban	159 (49.5)
Rural	162 (50.5)
**Living Conditions**	
Own House	133 (41.4)
Rented	188 (58.6)
**Family Conditions**	
Joint Family	204 (63.6)
Living Alone/No family members	117 (36.4)
**Monthly Income**	
<20,000	56 (17.4)
20–50k	111 (34.6)
50–100k	83 (25.9)
>100k	71 (22.1)
**Obesity**	
Yes	175 (54.5)
No	146 (45.5)
**Duration of T2DM Diagnosis**	
Less than 1 year	105 (32.7)
1 to 2 years	21 (6.5)
2 to 5 years	64 (19.9)
More than 5 years	131 (40.8)
**Family history of Diabetes**	
Yes	117 (55.1)
No	144 (44.9)
**Hypertension**	
Yes	149 (46.4)
No	172 (53.6)
**Dyslipidemia (high cholesterol, etc.)**	
Yes	160 (49.8)
No	161 (50.2)
**PCOS (for females)**	
Yes	86 (26.3)
No	80 (24.9)
Not applicable	155 (48.3)
**Renal Disease**	
Yes	146 (45.5)
No	175 (54.5)
**Cardiovascular Disease (CVD) History**	
Yes	158 (49.2)
No	163 (50.8)
**Non-Alcoholic Fatty Liver Disease**	
Yes	147 (45.8)
No	174 (54.2)
**Smoking Status**	
Never	113 (35.2)
Current	137 (42.7)
Ex Smoker	71 (22.1)
**Physical Activity (≥150 min/week?)**	
Yes	171 (53.3)
No	150 (46.7)
**Dietary Pattern**	
Healthy (low fat/sugar)	149 (46.4)
Unhealthy	172 (53.6)
**Adherence to given treatment plan**	
Yes	137 (42.7)
No	184 (57.3)

**Table 2 healthcare-13-02733-t002:** Socio-Demographic Predictors Influencing Diabetes Control: Findings from a Cross-Sectional Study Conducted in Community Pharmacies of Lahore, Pakistan (*n* =321).

Variables	Control of Diabetes Mellitus (N %)			Univariate Analysis		Multivariate Analysis		
	Control	Partially Control	Uncontrol	Crude OR (95% CI)	*p*–Value	Adjusted OR (95% CI)	*p*–Value	η^2^ (Effect Size)
**Gender**								
Male	57 (37.7)	60 (39.7)	34 (22.5)	Referent		Referent		
Female	65 (38.2)	62 (36.5)	43 (25.3)	0.843 (0.358–1.581)	0.784	0.743 (0.259–1.389)	0.967	-
**Age groups**								
18–40 years	38 (43.7)	29 (33.3)	20 (23.0)	Referent		Referent		
41–65 years	41 (35.0)	41 (35.0)	35 (29.9)	0.980 (0.587–1.038)	0.196	0.357 (0.337–1.397)	0.987	-
More than 65 years	43 (36.8)	52 (44.4)	22 (18.8)	0.123 (0.039–1.112)	0.056	0.363 (0.012–1.098)	0.097	-
**Occupation Type**								
Sedentary	50 (54.3)	31 (33.7)	11 (12.0)	Referent		Referent		
Active	23 (39.7)	20 (34.5)	15 (25.9)	2.228 (1.197–3.297)	0.021	1.099 (1.097–2.229)	0.061	-
Unemployed	14 (24.6)	28 (49.1)	15 (26.3)	2.397 (1.201–3.277)	0.001	1.208 (1.087–1.237)	** *0.047* **	0.01
Retired	35 (30.7)	43 (37.7)	36 (31.6)	2.397 (1.201–3.277)				-
**Education Level**								
No formal	27 (24.5)	43 (39.1)	40 (36.4)	Referent		Referent		
Primary	36 (31.9)	55 (48.7)	22 (19.5)	2.228 (1.391–3.098)	<0.001	1.307 (1.381–2.207)	** *0.006* **	0.06
Secondary	52 (65.0)	20 (25.0)	8 (10.0)	2.333 (1.961–3.987)	<0.001	1.317 (1.031–2.296)	** *0.001* **	0.07
Graduate	5 (50.0)	2 (20.0)	3 (30.0)	3.336 (1.271–3.127)	<0.001	2.338 (1.301–2.967)	** *0.001* **	0.06
Postgraduate	2 (25.0)	2 (25.0)	4 (50.0)	2.697 (1.361–4.207)	0.004	1.378 (1.871–3.037)	** *0.006* **	0.02
**Marital Status**								
Single	43 (32.1)	59 (44.0)	32 (23.9)	Referent		Referent		
Married	40 (44.9)	27 (30.3)	22 (24.7)	1.657 (1.089–1.987)	0.031	1.117 (1.361–1.207)	0.056	-
Divorced	32 (35.6)	36 (40.0)	22 (24.4)	1.597 (1.092–1.369)	0.045	1.027 (1.029–0.989)	0.052	-
Widowed	7 (87.5)	0 (0.0)	1 (12.5)	1.967 (1.669–1.557)	0.055	1.007 (0.989–0.087)	0.069	-
**Number of children**								
No child	31 (39.7)	32 (41.0)	15 (19.2)	Referent		Referent		
1 Child only	68 (36.6)	73 (39.2)	45 (24.2)	1.784 (1.289–1.995)	0.047	1.027 (1.189–0.955)	0.058	-
Less than 2	6 (30.0)	7 (35.0)	7 (35.0)	1.995 (1.681–1.956)	0.096	1.255 (1.870–1.323)	0.126	-
More than 2	17 (45.9)	10 (27.0)	10 (27.0)	2.622 (2.669–1.887)	0.098	1.607 (1.080–2.980)	0.459	-
**Living Conditions**								
Own House	30 (22.6)	65 (48.9)	38 (28.6)	Referent		Referent		
Rented	92 (48.9)	57 (30.3)	39 (20.7)	2.611 (2.089–1.367)	<0.001	1.602 (1.025–1.987)	** *0.045* **	0.01
**Residence**								
Urban	36 (22.6)	76 (47.8)	47 (29.6)	Referent		Referent		
Rural	86 (53.1)	46 (28.4)	30 (18.5)	1.982 (1.49–2.997)	<0.001	0.857 (1.119–0.955)	** *0.001* **	0.05
**Monthly Income (PKR)**								
<20,000	22 (39.3)	22 (39.3)	12 (24.4)	Referent		Referent		
20k–50k	47 (42.3)	43 (38.7)	21 (18.9)	2.617 (1.489–2.084)	0.323	1.257 (1.011–1.287)	0.498	-
50k–100k	25 (30.1)	36 (43.4)	22 (26.5)	2.347 (2.989–3.927)	0.597	1.557 (1.289–1.680)	0.985	-
>100k	28 (39.4)	21 (29.6)	22 (31.0)	2.611 (1.769–2.967)	0.458	1.612 (1.099–2.045)	0.896	-
**Obesity**								
Yes	42 (24.0)	83 (47.4)	50 (28.6)	Referent		Referent		
No	80 (54.8)	39 (26.7)	27 (18.5)	1.987 (1.589–2.907)	<0.001	1.057 (1.019–1.287)	** *0.006* **	0.02

Crude Odds Ratios (OR): Define the crude OR as unadjusted estimates representing the direct relationship between each predictor and the outcome, without controlling for other variables. Adjusted Odds Ratios (AOR): Specify that the AOR accounts for potential confounders by adjusting for covariates included in the model, providing a more accurate estimate of the relationship. Effect size was determined using Partial Eta Squared (η^2^). Based on Cohen’s classification, an effect size is considered small if 0.01 ≤ η^2^ ≤ 0.06, medium if 0.06 ≤ η^2^ ≤ 0.14, and large if η^2^ ≥ 0.14. Note: Outcome coded as Good Diabetes Control = 1 (reference = Poor Control). AOR > 1 indicates higher odds of good control; AOR < 1 indicates lower odds of good control. Note: To ensure transparency of all analyzed variables, both significant and non-significant predictors are reported in full. Statistically significant values (*p* < 0.05) are highlighted in bold for clarity.

**Table 3 healthcare-13-02733-t003:** Clinical Predictors Influencing Diabetes Control: Findings from a Cross-Sectional Study Conducted in Community Pharmacies of Lahore, Pakistan (*n* =321).

Variables	Control of Diabetes Mellitus (N %)			Univariate Analysis		Multivariate Analysis		
	Control	Partially Control	Uncontrol	Crude OR (95% CI)	*p*–Value	Adjusted OR (95% CI)	*p*–Value	η^2^ (Effect Size)
**Duration of T2DM Diagnosis**								
Less than 1 year	20 (19.0)	51 (48.6)	34 (32.4)	Referent		Referent		
1 to 2 years	5 (23.8)	9 (42.9)	7 (33.3)	1.251 (1.098–2.117)	0.041	0.957 (0.989–1.367)	0.052	-
2 to 5 years	18 (28.1)	35 (54.7)	11 (17.2)	2.457 (2.689–3.187)	0.035	1.217 (1.189–2.187)	** *0.047* **	0.02
More than 5 years	79 (60.3)	27 (20.6)	25 (19.1)	2.127 (1.989–2.977)	<0.001	1.277 (1.095–1.385)	** *0.008* **	0.04
**Family history of Diabetes**								
Yes	89 (50.3)	48 (27.1)	40 (22.6)	Referent		Referent		
No	33 (22.9)	74 (51.4)	37 (25.7)	2.967 (2.129–3.907)	<0.001	1.967 (1.689–3.287)	** *<0.001* **	0.09
**Hypertension**								
Yes	43 (28.9)	59 (39.6)	47 (31.5)	Referent		Referent		
No	79 (45.9)	63 (36.6)	30 (17.4)	2.367 (2.284–3.557)	0.002	1.857 (1.129–2.558)	0.055	-
**Dyslipidemia (High cholesterol)**								
Yes	52 (32.5)	62 (38.8)	46 (28.8)	Referent		Referent		
No	70 (43.5)	60 (37.3)	31 (19.3)	2.007 (1.281–3.027)	0.061	1.981 (1.381–2.007)	0.098	-
**PCOS (for females)**								
Yes	45 (52.3)	23 (26.7)	18 (20.9)	Referent		Referent		
No	18 (22.5)	38 (47.5)	24 (30.0)	2.115 (1.299–2.897)	0.003	1.787 (1.129–1.567)	** *0.039* **	0.01
Not Applicable	59 (38.1)	61 (39.4)	35 (22.6)	2.687 (2.181–1.587)	0.042	1.127 (1.089–1.557)	0.098	-
**Cardiovascular Disease (CVD) History**								
Yes	35 (22.2)	71 (44.9)	52 (32.9)	Referent		Referent		
No	87 (53.4)	51 (31.3)	25 (15.3)	2.257 (2.189–3.891)	<0.001	1.567 (1.220–2.677)	** *0.002* **	0.02
**Renal Diseas6**								
Yes	40 (27.4)	61 (41.8)	45 (30.8)	Referent		Referent		
No	82 (46.9)	61 (34.9)	32 (18.3)	2.447 (2.184–3.447)	0.001	1.981 (1.549–2.547)	** *0.035* **	0.01
**Non–Alcoholic Fatty Liver Disease**								
Yes	26 (17.7)	61 (41.5)	60 (40.8)	Referent		Referent		
No	96 (55.2)	61 (35.1)	77 (24.0)	2.987 (2.089–1.981)	<0.001	0.567 (0.218–0.941)	** *0.004* **	0.05
**Smoking Status**								
Never	30 (26.5)	43 (38.1)	40 (35.4)	Referent		Referent		
Current	36 (26.3)	66 (48.2)	35 (25.5)	2.307 (1.589–3.601)	<0.001	1.317 (0.914–1.649)	** *0.001* **	0.04
Ex–Smoker	56 (78.9)	13 (18.3)	2 (2.8)	2.367 (2.233–3.041)	<0.001	0.667 (0.255–1.041)	** *<0.001* **	0.08
**Physical Activity (≥150 min/week?)**								
Yes	86 (50.3)	44 (25.7)	41 (24.0)	Referent		Referent		
No	36 (24.0)	78 (52.0)	36 (24.0)	2.112 (2.031–3.551)	<0.001	1.387 (1.203–2.221)	** *<0.001* **	0.09
**Dietary Pattern**								
Healthy (low fat/sugar)	88 (59.1)	34 (22.8)	27 (18.1)	Referent		Referent		
Unhealthy	34 (19.8)	88 (51.2)	50 (29.1)	2.116 (2.023–3.089)	<0.001	1.317 (1.203–1.941)	** *<0.001* **	0.08
**Adherence to given treatment plan**								
Yes	102 (74.5)	21 (15.3)	14 (10.2)	Referent		Referent		
No	20 (10.9)	101 (54.9)	63 (34.2)	2.089 (1.833–2.841)	<0.001	1.287 (0.883–1.871)	** *<0.001* **	0.09

Effect size was determined using Partial Eta Squared (η^2^). Based on Cohen’s classification, an effect size is considered small if 0.01 ≤ η^2^ ≤ 0.06, medium if 0.06 ≤ η^2^ ≤ 0.14, and large if η^2^ ≥ 0.14. Note: To ensure transparency of all analyzed variables, both significant and non-significant predictors are reported in full. Statistically significant values (*p* < 0.05) are highlighted in bold for clarity.

## Data Availability

The original contributions presented in this study are included in the article. Further inquiries can be directed to the corresponding author.

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
