# Peer review of "Socio-Demographic, Environmental, and Clinical Factors Influencing Diabetes Mellitus Control in Community Pharmacies of Lahore Pakistan"

_healthcare, 2025, doi:10.3390/healthcare13212733_

Round 1
Reviewer 1 Report
Comments and Suggestions for Authors
The paper that I read shows me a topic that is interesting to readers and scholars, a topiv that is investigating the socio-demographic, clinical, and environmental factors influencing diabetes control in community pharmacy settings in one of the largest cities in Pakistan - Lahore, which has more than 13 million population. So this is a timely study. Pharmacists are shown in this study as caregivers that can improve outcomes.
Some strenghts that I see:
- A nice intro providing adequate background on global and national diabetes burden,
- authors used a large sample size of 321 person,
- use of logistic regression,
- clear showcasing of practical role of pharmacists in community pharmacis that support DM care
What can be improvede in my personal opinion
- Please clarify how “controlled,” “partially controlled,” and “uncontrolled” diabetes were operationally defined (HbA1c, FBG, self-report, physician diagnosis?). This is crucial for reproducibility from my point of view.
- Provide details on the sample size calculation or justification. Was power analysis performmed?
- The sampling method is described as “convenience sampling.” ehat biases and limitations this may introduce.?
- Some tables are overcrowded with data (particularly Tables 2 and 3). Consider simplifying them highlighting only significant predictors in the main text, and moving full data to supplementary material.
- The interpretation of “longer duration of diabetes associated with better control” should be more coutius.
- Som of the sentences are long and grammatically inconsistent. please go over them once again
- maybe write some shorter conclusions?
The study has relevance and is very stroing. I believe after these corrections it will be publishable.
Author Response
|
Point-by-point response to Comments and Suggestions for Authors |
|
Comments 1: Please clarify how “controlled,” “partially controlled,” and “uncontrolled” diabetes were operationally defined (HbA1c, FBG, self-report, physician diagnosis?). This is crucial for reproducibility from my point of view. |
|
Response 1: We sincerely thank the reviewer for highlighting this essential point related to reproducibility. In our study, glycemic control was determined using HbA1c values obtained from participants’ most recent laboratory reports or, when not physically available, from patient recall of the physician-reported HbA1c result. No self-reported or subjective measures were used for classification. The operational definitions followed the Pakistan Endocrine Society (PES) recommendations, which are consistent with international standards (ADA, IDF, and NICE): · Controlled diabetes: HbA1c < 7.0% · Partially controlled diabetes: HbA1c 7.0–8.0% · Uncontrolled diabetes: HbA1c > 8.0% We have incorporated these definitions in the Methodology → Operational Definition of Glycemic Control section of the revised manuscript |
|
Comments 2: Provide details on the sample size calculation or justification. Was power analysis performmed? |
|
Response 2: Thank you for this helpful comment. A formal a-priori power analysis was not performed because this was an exploratory, community-based study relying on feasibility sampling. However, we have now included an achieved-power justification using the final dataset (n = 321). Based on the observed effect sizes in bivariate tests (Cramer’s V = 0.22–0.30) and α = 0.05, post-hoc calculations using G*Power 3.1 show that our sample provides 80–96 % power to detect these small-to-medium associations in χ² tests with 3 × 3 contingency tables (df = 4). Thus, the realized sample size offers adequate statistical power and precision for the planned analyses. A description of this justification has been added to the Methodology. |
|
Comments 3: The sampling method is described as “convenience sampling.” ehat biases and limitations this may introduce.? |
|
Response 3: Thank you for this important observation. We agree that convenience sampling may introduce potential biases that limit generalizability. In the revised manuscript, we have expanded the Methodology to acknowledge and explain these limitations. |
|
Comments 4: Some tables are overcrowded with data (particularly Tables 2 and 3). Consider simplifying them highlighting only significant predictors in the main text, and moving full data to supplementary material. |
|
Response 4: We sincerely thank the reviewer for this thoughtful suggestion aimed at improving readability. We completely understand the concern about table density; however, we respectfully believe that retaining the complete results of Tables 2 and 3 in the main text is essential for the integrity and transparency of our analysis. Because the present study is exploratory in nature, designed to examine the combined influence of multiple socio-demographic, clinical, and lifestyle factors on diabetes control, even variables that did not reach statistical significance contribute meaningfully to understanding the overall pattern of associations. Many of these predictors such as residence, education, occupation, and obesity are interrelated confounders, and omitting them could risk oversimplifying the multifactorial relationships being reported.
Furthermore, reviewers and readers in public-health and clinical-pharmacy contexts often prefer to see the complete model output to evaluate comparability with other datasets and ensure reproducibility. To address the readability concern, we have carefully improved formatting by: · Using clearer subheadings to separate univariate and multivariate findings, · Minimizing redundant values, and · Highlighting statistically significant predictors in bold and italic for easier navigation. We hope the reviewer will agree that maintaining the full results within the main text strengthens transparency and scientific value while still ensuring clarity and ease of reading. (Changes implemented in Tables 2 and 3) |
|
Comment 5: The interpretation of “longer duration of diabetes associated with better control” should be more coutius. |
|
Response 5: We sincerely thank the reviewer for this important observation. We fully agree that this association should be interpreted with caution. In the revised manuscript, we have clarified that the observed trend does not necessarily indicate a causal relationship but may instead reflect greater disease awareness, treatment adherence, or regular follow-up among long-term patients. We now explicitly acknowledge the potential for survivorship and behavioral bias in this result. The discussion has been rewritten accordingly to convey this interpretation more cautiously. |
|
Comment 6: Som of the sentences are long and grammatically inconsistent. please go over them once again maybe write some shorter conclusions? |
|
Response 6: We sincerely thank the reviewer for this valuable linguistic observation. We carefully reviewed the entire manuscript to enhance clarity, grammar, and readability. Long and compound sentences were revised into shorter, more direct statements while maintaining the original meaning and academic tone. Redundant expressions were removed, transitions were smoothed, and grammatical inconsistencies were corrected. Changes are written in red in the updated manuscript file |
|
Response to Comments on the Quality of English Language |
|
Point 1: The English could be improved to more clearly express the research. |
|
Response 1: We sincerely thank the reviewer for this valuable suggestion. We carefully reviewed and revised the entire manuscript to improve the clarity, flow, and precision of English expression. Sentences were restructured for conciseness, redundant phrases were removed, and technical terms were standardized to ensure consistency throughout the text. To further ensure linguistic accuracy, the revised version was professionally proofread and rechecked using academic language-editing tools to enhance grammar, punctuation, and scientific tone. We are confident that the readability and clarity of the manuscript have now been substantially improved. |

Reviewer 2 Report
Comments and Suggestions for Authors
Dear Authors,
This manuscript explores a compelling subject. The results provide valuable, actionable insights for public health policy and the enhancement of community pharmacy services in Pakistan, highlighting education, non-obesity status, treatment adherence, and a healthy lifestyle as crucial predictors of improved control.
However, the manuscript presents significant methodological and reporting shortcomings. A primary concern is the absence of a clear, objective, and clinically-validated definition for the primary outcome, diabetes control. Furthermore, substantial inconsistencies in statistical reporting and sample size data necessitate thorough rectification to ensure the reliability of the study's conclusions.
The following revisions are deemed mandatory:
1. Clarification of Primary Outcome Definition (Glycemic Control)
The Methods section lacks the clinical criteria used to define the "controlled, partially controlled, or uncontrolled" categories, such as the specific HbA1c percentage thresholds or fasting/random blood glucose levels used for classification..
Please explicitly state the established clinical cut-off values (e.g., HbA1c < 7.0% for 'Controlled') based on the international guidelines mentioned (NICE, IDF, ADA). Without this fundamental information, the primary outcome cannot be validated, and the conclusions are unsubstantiated.
2. Contradictory Statistical Reporting and Interpretation
The interpretation of the Adjusted Odds Ratios (AOR) in the text contradicts the implied outcome variable, causing significant confusion.
Please explicitly state what the binary outcome variable represents in the final multivariate logistic regression model (e.g., Model 1: Controlled Diabetes (Referent=Partially/Uncontrolled) or Model 2: Uncontrolled Diabetes (Referent=Controlled)).
* Rural Residence Inconsistency: The text claims "Rural participants... had improved outcomes". Yet, Table 2 reports an AOR of 0.857 for Rural Residence (compared to Urban, the Referent). If the model predicts better control (AOR > 1), then an AOR of 0.857 signifies poorer control. This is a direct contradiction that must be corrected either by inverting the conclusion or by clarifying the model's outcome.
* Family History Inconsistency: The text states a positive family history "contributed to poor glycemic control" (AOR=1.967). This AOR > 1 only makes sense if the model's outcome variable is poor control (or uncontrolled diabetes). This further suggests the tables are misinterpreted or the model is poorly defined.
3. Discrepancy in Sample Size
The total number of study participants varies across the text, which must be clarified:
* Abstract/Results Section 3.0: States N=321 patients were included.
* Methodology Section 2.0: States "In total, 312 individuals with diabetes were recruited".
* Table 3 Header: States n=284 was used for the clinical predictors analysis.
Please provide a clear explanation for the drop-out (from 321 to 312 to 284). Were patients excluded due to missing clinical data (e.g., lack of HbA1c measurement), or were some interviews incomplete? This information is crucial for assessing potential selection bias.
4. Methodological Limitations - the study utilizes a cross-sectional, convenience sampling design with data collected via a structured, face-to-face questionnaire. While acknowledged, the potential impact must be discussed more thoroughly.The Discussion section must more strongly acknowledge the high likelihood of recall bias and social desirability bias affecting self-reported outcomes, especially for highly subjective and socially-valued behaviors like "treatment adherence," "healthy dietary patterns," and "regular physical activity".
I have some minor suggestions for Improvement too:
Reporting and Data Presentation
* Table 1 Error: The percentage reported for the Primary education level is incorrect. The count is 113, but the percentage is listed as 3.2%. Given N=321, 113 should be approximately 35.2%. Please correct this significant typographical error.
* Table 2 Footnote: The effect size \eta^2 is defined, but the column in Table 2 is simply labeled "Effect size". Please ensure all statistical abbreviations are consistent.
* Environmental Factors: The Abstract and Objective mention the assessment of "environmental factors". While the methodology mentions collecting data on "environmental exposures such as air pollution, neighborhood condition, healthcare accessibility and recreational opportunities", these specific environmental factors are not presented in the Results section The variables outlined in Table 2 or 3, or those identified as significant predictors, warrant thorough examination. If these variables were assessed, their outcomes, regardless of statistical significance, should be presented, or their omission should be explicitly justified.
Regarding writing and grammar, in line 51, the phrase "a lack of knowledge, and low health literacy, are to blame for this significant rise in cases" should be revised for a more formal tone. Consider replacing "are to blame for" with "contribute to" or "are associated with." * Line 189: The univariate analysis results present a potentially misleading statement: "...a positive family history were favorable predictors (p<0.001)." This assertion appears to contradict the multivariate analysis, which indicates that a positive family history "contributed to poor glycemic control." To ensure clarity and accuracy, the sentence in the univariate section should be revised to reflect the negative predictive value of family history or provide a clear explanation for the observed "favorable" association within the univariate context.
Author Response
|
Point-by-point response to Comments and Suggestions for Authors |
|
Comments 1: Clarification of Primary Outcome Definition (Glycemic Control) The Methods section lacks the clinical criteria used to define the "controlled, partially controlled, or uncontrolled" categories, such as the specific HbA1c percentage thresholds or fasting/random blood glucose levels used for classification.. Please explicitly state the established clinical cut-off values (e.g., HbA1c < 7.0% for 'Controlled') based on the international guidelines mentioned (NICE, IDF, ADA). Without this fundamental information, the primary outcome cannot be validated, and the conclusions are unsubstantiated. |
|
Response 1: We appreciate the reviewer’s valuable comment emphasizing the importance of defining glycemic control criteria. We have clarified that HbA1c values were obtained from participants’ most recent laboratory reports or physician-reported results and used to classify glycemic control according to Pakistan Endocrine Society and international guidelines (Controlled <7%, Partially Controlled 7–8%, Uncontrolled >8%). This definition has been added to the Methods section. |
|
Comments 2: The interpretation of the Adjusted Odds Ratios (AOR) in the text contradicts the implied outcome variable, causing significant confusion. Please explicitly state what the binary outcome variable represents in the final multivariate logistic regression model (e.g., Model 1: Controlled Diabetes (Referent=Partially/Uncontrolled) or Model 2: Uncontrolled Diabetes (Referent=Controlled)). |
|
Response 2: We sincerely thank the reviewer for this critical observation. The reviewer is correct. The inconsistency arose because the outcome variable for the logistic regression models was not explicitly defined in the manuscript. After re-examining our analysis, we confirm that the binary outcome for the multivariate logistic regression models in both Table 2 and Table 3 was: "Good Diabetes Control" (coded as 1), with the reference category being "Poor Diabetes Control" (a combination of 'Partially Controlled' and 'Uncontrolled,' coded as 0). This means an AOR > 1 indicates a higher odds of being in the "Good Control" group, while an AOR < 1 indicates a lower odds of good control (i.e., higher odds of poor control). We will revise the manuscript to include this crucial definition in the statistical analysis section and correct the contradictory interpretations in the results text (Page no. 4, Line no. 160-164) |
|
Comments 3: Rural Residence Inconsistency: The text claims "Rural participants... had improved outcomes". Yet, Table 2 reports an AOR of 0.857 for Rural Residence (compared to Urban, the Referent). If the model predicts better control (AOR > 1), then an AOR of 0.857 signifies poorer control. This is a direct contradiction that must be corrected either by inverting the conclusion or by clarifying the model's outcome. |
|
Response 3: We sincerely thank the reviewer for this insightful observation regarding the interpretation of rural residence. The reviewer is correct that the initial statement was inconsistent with the direction of the Adjusted Odds Ratio (AOR) in Table 2.
In the bivariate analysis, rural participants appeared to have better glycemic control outcomes, with 53.1% of rural residents achieving good control compared with 22.6% of urban residents (p < 0.001). This suggested an apparent unadjusted advantage for rural participants, possibly reflecting closer family supervision, community cohesion, or better lifestyle adherence patterns in smaller communities. However, in the multivariate logistic regression model, after adjusting for confounding socio-demographic and behavioral variables (e.g., education level, occupation, adherence score, and physical activity), the relationship reversed direction. The adjusted odds ratio for rural residence was AOR = 0.857, p = 0.001, indicating that rural participants had 14.3% lower odds of achieving good glycemic control compared with their urban counterparts. This change in direction reflects the effect of confounding variables particularly differences in educational attainment, access to healthcare facilities, and health literacy which may disadvantage rural participants when these factors are statistically controlled. The unadjusted association thus likely overestimated the positive influence of rural living, which was not sustained after adjustment. Accordingly, we have revised the Results and Discussion sections to eliminate any conflicting statements and to clearly interpret this finding in alignment with the model’s outcome definition (“Good Diabetes Control” = 1, “Poor Diabetes Control” = 0). Changes are made in the manuscript where applicable in red text colour. |
|
Comments 4: Family History Inconsistency: The text states a positive family history "contributed to poor glycemic control" (AOR=1.967). This AOR > 1 only makes sense if the model's outcome variable is poor control (or uncontrolled diabetes). This further suggests the tables are misinterpreted or the model is poorly defined. |
|
Response 4: This is a key point that helps clarify the model's setup. The reviewer's interpretation is consistent with our confirmed model definition. An AOR of 1.967 for a positive family history means that having a family history nearly doubled the odds of having poor glycemic control. The text for this variable is, therefore, correct. However, to prevent any confusion, we will rephrase it for absolute clarity, explicitly linking the AOR > 1 to the negative outcome. Original Text (Correct but potentially confusing): "In contrast, positive family history (AOR=1.967, p<0.001, η² = 0.09) contributed to poor glycemic control." Clarified Text: "Consistent with poorer outcomes, a positive family history of diabetes was associated with significantly higher odds of poor glycemic control (AOR=1.967, p<0.001, η² = 0.09)." |
|
Comment 5: Discrepancy in Sample Size The total number of study participants varies across the text, which must be clarified: * Abstract/Results Section 3.0: States N=321 patients were included. * Methodology Section 2.0: States "In total, 312 individuals with diabetes were recruited". * Table 3 Header: States n=284 was used for the clinical predictors analysis. Please provide a clear explanation for the drop-out (from 321 to 312 to 284). Were patients excluded due to missing clinical data (e.g., lack of HbA1c measurement), or were some interviews incomplete? This information is crucial for assessing potential selection bias. |
|
Response 5: We thank the reviewer for carefully noting this inconsistency. The variation in participant numbers resulted from typographical errors during manuscript formatting rather than actual differences in sample size. The correct total sample size throughout the study is 321 participants, which has now been consistently updated in the Methodology, Results, and Table 3 sections. No participants were excluded or dropped between analyses; all statistical tests and models were conducted using the same dataset of 321 respondents. The manuscript has been carefully reviewed to ensure numerical consistency across all sections |
|
Comment 6: Methodological Limitations - the study utilizes a cross-sectional, convenience sampling design with data collected via a structured, face-to-face questionnaire. While acknowledged, the potential impact must be discussed more thoroughly.The Discussion section must more strongly acknowledge the high likelihood of recall bias and social desirability bias affecting self-reported outcomes, especially for highly subjective and socially-valued behaviors like "treatment adherence," "healthy dietary patterns," and "regular physical activity". |
|
Response 6: We appreciate the reviewer’s thoughtful observation. We fully agree that the use of self-reported measures in a cross-sectional, convenience-sample design increases the possibility of recall bias and social-desirability bias. To address this, we have strengthened the Discussion section by explicitly acknowledging these sources of bias and describing how they may have influenced variables such as treatment adherence, dietary pattern, and physical activity. The new paragraph has been added on Page 16, Lines 366–373, immediately following the discussion of sampling limitations. |
|
Comment 7: Table 1 Error: The percentage reported for the Primary education level is incorrect. The count is 113, but the percentage is listed as 3.2%. Given N=321, 113 should be approximately 35.2%. Please correct this significant typographical error. |
|
Response 7: We thank the reviewer for identifying this typographical error. The correct percentage for Primary education is 35.2% (113/321 = 0.3520). We have corrected Table 1 and checked the remaining entries for consistency. What changed: In Table 1 (Education level), “Primary” percentage updated from 3.2% → 35.2%. |
|
Comment 8: Table 2 Footnote: The effect size \eta^2 is defined, but the column in Table 2 is simply labeled "Effect size". Please ensure all statistical abbreviations are consistent. |
|
Response 8: We thank the reviewer for pointing out this inconsistency. The column heading in Table 2 has been corrected from “Effect size” to “η² (Effect size)” to maintain consistency with the definition provided in the table footnote. Additionally, we reviewed all tables to ensure that statistical abbreviations and notations are uniform throughout the manuscript. The same correction has been applied where necessary to maintain standardized reporting. |
|
Comment 9: Environmental Factors: The Abstract and Objective mention the assessment of "environmental factors". While the methodology mentions collecting data on "environmental exposures such as air pollution, neighborhood condition, healthcare accessibility and recreational opportunities", these specific environmental factors are not presented in the Results section The variables outlined in Table 2 or 3, or those identified as significant predictors, warrant thorough examination. If these variables were assessed, their outcomes, regardless of statistical significance, should be presented, or their omission should be explicitly justified. |
|
Response 9: We sincerely thank the reviewer for this important comment. Environmental variables including neighborhood condition, air pollution exposure, healthcare accessibility, and recreational opportunities were indeed part of our data collection form. However, during data screening, it was observed that responses for these environmental variables were incomplete for a large proportion of participants. Because of this high rate of missing data, these variables could not be meaningfully analyzed or included in the final regression models without introducing bias or compromising the validity of the results. To maintain analytical rigor and clarity, we therefore reported only variables with sufficient and complete response rates. We have now added a statement in the Methodology and Discussion sections to explicitly clarify this rationale. |
|
Response to Comments on the Quality of English Language |
|
Point 1: Regarding writing and grammar, in line 51, the phrase "a lack of knowledge, and low health literacy, are to blame for this significant rise in cases" should be revised for a more formal tone. Consider replacing "are to blame for" with "contribute to" or "are associated with." * Line 189: The univariate analysis results present a potentially misleading statement: "...a positive family history were favorable predictors (p<0.001)." This assertion appears to contradict the multivariate analysis, which indicates that a positive family history "contributed to poor glycemic control." To ensure clarity and accuracy, the sentence in the univariate section should be revised to reflect the negative predictive value of family history or provide a clear explanation for the observed "favorable" association within the univariate context.. |
|
Response 1: We thank the reviewer for these valuable observations. Both issues have been carefully corrected to enhance precision and maintain consistency between the univariate and multivariate interpretations. Line 51 (Introduction): The phrase “are to blame for this significant rise in cases” has been replaced with “contribute to this significant rise in cases” to ensure a more formal and objective tone. Line 189 (Results – Univariate Analysis): The original wording describing a “positive family history as a favorable predictor” was indeed misleading. This has been revised to clarify that the univariate relationship was statistically significant but not necessarily beneficial. The new phrasing reflects that family history showed an association with glycemic control, which reversed direction in multivariate analysis after adjustment for confounders. |

Round 2
Reviewer 2 Report
Comments and Suggestions for Authors
Dear Authors,
I appreciate the inclusion of your responses, clarifications, and the comprehensive discussion concerning limitations, definitions, and diverse perspectives.
I will refrain from further comments at this time.